# Pediatric Hepatocellular Adenomas: What Is Known and What Is New?

**DOI:** 10.3390/cancers15194790

**Published:** 2023-09-29

**Authors:** Andres F. Espinoza, Sanjeev A. Vasudevan, Prakash M. Masand, Dolores H. Lòpez-Terrada, Kalyani R. Patel

**Affiliations:** 1Divisions of Pediatric Surgery and Surgical Research, Michael E. DeBakey Department of Surgery, Pediatric Surgical Oncology Laboratory, Texas Children’s Surgical Oncology Program, Texas Children’s Liver Tumor Program, Dan L. Duncan Cancer Center, Baylor College of Medicine, Houston, TX 77030, USA; andyespinoza.espinoza@bcm.edu (A.F.E.); savasude@texaschildrens.org (S.A.V.); 2Department of Radiology, Baylor College of Medicine and Texas Children’s Hospital, Houston, TX 77030, USA; pmmasand@texaschildrens.org; 3Department of Pathology and Immunology, Baylor College of Medicine and Texas Children’s Hospital, Houston, TX 77030, USA; dhterrad@texaschildrens.org; 4Abercrombie B1, Texas Children’s Hospital, Baylor College of Medicine, 6621 Fannin St., Houston, TX 77030, USA

**Keywords:** liver, adenoma, pediatric

## Abstract

**Simple Summary:**

Pediatric hepatocellular adenoma (HCA) is a rare tumor and often observed in abnormal livers. Despite advances in adults, there is little progress in understanding the origin and management of pediatric HCAs. Evidence of multifocality, malignant transformation and rupture in a subset of patients warrant a thorough understanding of this rare entity to guide optimum management. We present a concise review of the current literature on HCA, with a focus and comparative assessment on pediatric patients.

**Abstract:**

Current understanding and classification of pediatric hepatocellular adenomas (HCA) are largely based on adult data. HCAs are rare in children and, unlike in adults, are often seen in the context of syndromes or abnormal background liver. Attempts to apply the adult classification to pediatric tumors have led to several “unclassifiable” lesions. Although typically considered benign, few can show atypical features and those with beta-catenin mutations have a risk for malignant transformation. Small lesions can be monitored while larger (>5.0 cm) lesions are excised due to symptoms or risk of bleeding/rupture, etc. Management depends on gender, age, underlying liver disease, multifocality, size of lesion, histologic subtype and presence of mutation, if any. In this review, we summarize the data on pediatric HCAs and highlight our experience with their diagnosis and management.

## 1. Introduction

Pediatric hepatic adenoma or hepatocellular adenoma (HCA) is a rare liver tumor that comprises approximately 4% of all liver lesions in children [1]. While initially described as a benign liver lesion arising in normal adult livers, it is commonly seen in children with genetic disorders or abnormal livers [2]. Diagnosis and management require a multidisciplinary approach comprising radiologists, pathologists, surgeons, hepatologists, and oncologists. Considerations include choice of imaging and pathology tools for establishing the diagnosis, evaluating the risk of surgical versus conservative interventions, strategies for surveillance, and post-operative follow up to ensure optimum short- and long-term outcomes. Although biologically benign, HCAs can have high morbidity due to multifocal or large lesions, rupture, or malignant transformation. Malignant transformation can result in hepatocellular carcinoma and rarely hepatoblastoma [1,2,3,4]. General understanding of pediatric HCAs is limited given the rarity of this lesion. The majority of studies are single center with limited patients. Here, we present a concise review of pediatric HCAs including recent advances and potential future directions for the management of these rare tumors.

## 2. Risk Factors and Epidemiology

In adults, HCAs have an incidence of 4 per 100,000 patients and commonly arise from normal livers [5,6]. The normalcy of the background liver is often used in the diagnostic criteria for adult HCAs by pathologists. In contrast, HCAs in children have an incidence of approximately 1 per 1 million patients and commonly arise in those with congenital or pre-existing conditions affecting the liver [7,8]. In adults, obesity, alcoholism, and elevated estrogen levels driven by oral contraceptive pills (OCPs) have been the strongest risk factors for the development of HCAs [9]. Interestingly, pediatric disorders that drive similar hormonal dysregulation such as polycystic ovarian syndrome (PCOS), Klinefelter syndrome, and sex hormone-producing tumors have been associated with increased risk for HCAs [5,6,9]. Other genetic disorders such as glycogen storage diseases, galatosemia, ornithine transcarbamylase deficiency [10], immunodeficiency disorders, congenital hepatic fibrosis [11], HNF1A-maturity onset diabetes of young (previously termed MODY3) [12], Fanconi anemia, and rare conditions such as Wolf–Hirschhorn syndrome [13] have been associated with HCAs in children [1,14]. Glycogen storage diseases, in particular types I and III, have been shown to have a younger incidence of HCAs, multifocal tumors, and a higher risk of rupture compared to adenomas in the general pediatric population [15]. Among the acquired conditions, pediatric HCAs have been reported with the Fontan procedure [16], secondary to androgen therapy prior to hematopoietic stem cell transplantation [17], as a late complication of prior cancer therapy [18], and as a potential complication of oxcarbazepine therapy for seizure disorder [19]. All age groups are affected in children, including rare cases of prenatally detected tumors [20]. Unlike adults with strong female predominance, HCAs are seen equally in pre-pubertal girls and boys, but show female predominance in post-pubertal children [1,7,13]. (Table 1).

## 3. Presentation and Clinical Features

The majority of HCAs lack symptoms and are incidentally diagnosed [21,22]. Patients can also present with vague abdominal pain that appears to be non-positional, have no specific modifying factors, and can spontaneously increase or decrease in severity [22,23]. Other presentations can mimic acute appendicitis, malrotation, kidney stones, and gastroenteritis [21,22]. Some patients may present with obstructive symptoms such as nausea, emesis, and abdominal pain [21,22]. In contrast to adults, a significant subset of pediatric HCAs arise in the context of genetic disorders or other acquired conditions and their HCAs are more likely to be diagnosed incidentally during the surveillance imaging. Some asymptomatic cases are detected based upon suspicion by a caretaker or a family member [23]. In contrast, ruptured HCAs, can have a wide range of symptoms from vague abdominal pain to hypotensive crisis, peritonitis, and shock. Approximately 15% of patients that have HCA > 5.0 cm have been shown to present with hemorrhage as their presenting symptom [22]. 

Clinically unstable patients are usually triaged to the operating room, intensive care unit (ICU), or interventional radiology for stabilization. While the latter requires immediate intervention, they both require establishing the diagnosis via a stepwise process. The differential diagnosis for a pediatric liver mass includes hepatoblastoma, hepatocellular carcinoma, focal nodular hyperplasia, benign hemangioma, hepatic cyst, and HCA.

## 4. Imaging

Ultrasound (US) examination of the right upper quadrant with or without Doppler is the most common initial modality for diagnosis and shows solitary or multiple discrete echogenic mass lesions [23]. The appearance can be heterogeneous with intra-tumoral degeneration and hemorrhage. On doppler US, HCAs can show centrally located vessels with continuous venous waveforms [6,21,22,23]. Recent studies have found that contrast-enhanced US can help differentiating an HCA from focal nodular hyperplasia (FNH), as HCAs show centripetal arterial flow [6] While some evidence exists that contrast-enhanced US has similar specificity as contrast-enhanced MRI, other have demonstrated that the use of contrast-enhanced US can be used as an adjunct to improve the sensitivity of differentiating HCA from other liver masses [6].

MRI is considered the gold standard for diagnosis in children with HCAs, with MR imaging features well described in the literature. Lesions can be heterogenous due to internal fat, hemorrhage, necrosis, and variable vascularity. T1-weighted imaging tends to show high signal with intra-tumoral hemorrhage [6,24,25] (Figure 1A,B). MR imaging features of the various subtypes are well described, including hepatocyte nuclear factor-1-alpha (HNF-1α)-mutated HCA (H-HCA), β-catenin-mutated HCA (β-HCA), inflammatory HCA (I-HCA), and unclassified HCA (U-HCA). It is recommended that MR imaging be performed utilizing a hepatocyte-specific contrast agent, which are predominantly excreted via the biliary system. Almost 78% of H-HCAs exhibit diffuse signal loss on out-of-phase imaging because of intracellular fat content with a specificity of 100%) [25]. They can have variable signal on T2-weighted imaging, and show moderate enhancement in the arterial phase with no retention of hepatocyte-specific contrast on delayed phase imaging. I-HCAs are typically T2 hyperintense, especially peripherally (atoll sign) due to dilated sinusoids, as shown in Figure 1C [25]. I-HCAs exhibit avid arterial phase enhancement which persists on the portal venous and delayed phase images. β-HCAs do not have specific imaging features but can show intense arterial phase enhancement with subsequent washout [26]. While FNHs can be difficult to differentiate from HCAs on MRI, HCAs usually lack the central scar that is characteristic for FNH [6,14,16]. It is also important to monitor the progression of these tumors as early washout with arterial enhancement is characteristic of malignant transformation [6,22,24].

While US and MRI are the currently recommended imaging modalities for detection and monitoring of pediatric HCA, we recognize that several institutions worldwide may still rely on computed tomography (CT). The utility of CT is valid in situations where there is a lack of other imaging tools. Regardless, MRI and US should be utilized when available. The downside with CT is the high radiation dose imparted, especially when multiphase CT imaging is performed for the evaluation of suspected hepatic adenoma. On CT, HCA appear without central scar and can show a heterogenous appearance given hemorrhage in the tumor [13] Additionally, multifocal tumor burden may be hard to assess given the poor soft tissue contrast resolution, especially when a single post-contrast phase is obtained.

Female patients or patients that are poor surgical candidates with asymptomatic HCAs less than 5.0 cm are recommended to have surveillance MRI at 6 months intervals, for the first year [1,2,5,7,22]. If the tumor does not increase in size, MRIs can be extended to yearly surveillance [12,22]. Previous observational studies have shown that adult patients with HCAs greater than 5.0 cm, less than 20 percent had reduction in size after six months, one fourth showed reduction at 1 year, and 60 percent had resolution after 2 years [11,24]. Thus, accurate evaluation of size and close monitoring is warranted but is yet to be defined in the pediatric population. Hence, the management for HCAs based on size is determined by individual institutional practice guidelines, and sometimes family preferences.

## 5. Histopathology, Genetics and Classification

On histopathologic examination, HCAs are well-circumscribed, un-encapsulated or partially/thinly capsulated tumors with uniform cut surface. Large or long-standing tumors can show secondary changes such as intra-tumoral hemorrhage or infarction (Figure 2A). They are composed of proliferation of uniform, bland and benign appearing hepatocytes. Tumor cells are usually larger than the background liver and often contain excess glycogen or fat (Figure 2B). There is absent or minimal cytologic atypia and pleomorphism. Mitotic figures are usually absent. Intra-tumor growth pattern is trabecular, closely mimicking normal liver architecture with slightly wider, 2–3 cell thick hepatic cords. Areas of compressed sinusoids may appear sheet-like but definitive solid or acinar architecture or small cell change are not seen. Some lesions can show dilated sinusoids and atrophic cords with a peliosis-like appearance (Figure 2C,D). Nuclei are central, round, with smooth contours, vesicular chromatin, low nucleus to cytoplasm ratio and absent nucleoli. Older lesions with secondary changes such as hemorrhage and infarction can show some large, hyperchromatic, pleomorphic nuclei resembling large cell change (Figure 3A–D). Cytoplasm is abundant, often pale and can show pigment such as bile, lipofuscin, Dubin-Johnson pigment and rarely Mallory–Denk bodies. Tumors are typically vascular with small to medium sized muscular arteries traversing through the lesion (Figure 2F). These arteries can be accompanied by delicate stroma or fibrous tissue; however, tumoral fibrosis is not seen. Some tumors can show mixed inflammatory infiltrate composed of lymphocytes, histiocytes, plasma cells, neutrophils and rarely eosinophils within the hepatocytes or along the stroma of the traversing arteries (Figure 2E). Rarely, intra-tumoral granulomas have been noted. Tumors are devoid of portal tracts and bile ducts, but some entrapped structures can be seen along the periphery. HCAs lack central scar and bile ducts of focal nodular hyperplasia (FNH). A well-developed, intact reticulin framework is seen within the lesion. CD34 is typically patchy and highlights the vascular structures and nearby sinusoids. In a resection sample, it is rarely diffuse but small biopsies can show diffuse CD34 expression in some fragments and should not be confused with a dysplastic nodule or HCC (Figure 4D–F). Tumors are typically (except those with beta-catenin mutations as described below) negative for Glutamine synthetase, Glypican-3 and show membranous reactivity for beta catenin (Figure 4A–C). Adult HCAs usually arise in normal background liver. However, a recent multi-institutional series involving 31 children showed that 54% were associated with various syndromes characterized by minor (steatosis) or significant (cirrhosis) abnormalities in the background liver [27]. Few reports from adult patients have also described HCAs arising in alcoholic and hepatitis B-related cirrhosis [28,29].

Largely based on adult studies, HCAs are classified using histomorphology, immunophenotypic features, and molecular findings into the following subtypes:

**Inflammatory adenomas (i-HCA)**: These are the most common HCAs in adults and post-pubescent children [27,30]. They are typically seen in females and in those with high body mass index, and/or alcohol consumption [31,32]. They are also described in glycogen storage disease type 1 and primary sclerosing cholangitis [33,34]. Microscopically, they show dilated sinusoids and atrophic hepatic cords with a peliosis-like pattern. Varying degree of mononuclear inflammatory infiltrate can be seen along the vascular septa, along with ductular proliferation. Historically, they were classified as telangiectatic focal nodular hyperplasia. In early 2000, Paradis and Bioulac-Sage showed that telangiectatic focal nodular hyperplasia are monoclonal neoplasms that display molecular pattern closer to HCAs than FNH, since then these are classified as i-HCAs [35,36]. Tumor cells are positive for serum amyloid A (SAA) and C-reactive protein (CRP) by immunohistochemistry. They are typically negative for Glutamine synthetase and Glypican-3. They are characterized by somatic, gain of function mutations in the IL-6 signal transducer gene (*IL6ST*) which encodes for glycoprotein-130 (gp130), which is a component of the IL6 receptor [29]. This activation of IL6 promotes the STAT3 signaling pathway and induces an acute phase inflammatory response within the tumor, which is manifested by the expression of SAA and CRP. Somatic mutations of the *IL6ST* are seen in approximately 60% of inflammatory adenomas. Non-mutated inflammatory adenomas show similar inflammatory expression profiles including SAA and CRP reactivity by IHC, but the mechanisms are not clear [37]. There is no consensus on the requirement of SAA or CRP positivity or a specific mutation to diagnose an inflammatory HCA. Most cases are diagnosed on morphology alone. Cases without an inflammatory component or peliosis but with positive staining for SAA or CRP are particularly challenging and remain unexplained. Approximately 10% of inflammatory adenomas harbor mutations in the beta catenin gene (*CTNNB1*) giving rise to Wnt/beta-catenin pathway activation. This subset, in addition to SAA and CRP, can show nuclear reactivity for beta-catenin by immunohistochemistry and will have increased risk for malignant transformation.

**HNF1-alpha mutated adenomas (H-HCA)**: This is the second most common HCA subtype in adults and children [19,22]. Microscopically, they show moderate to severe lesional steatosis and absence of peliosis and inflammation. Lesional cells are negative for SAA and CRP and show loss of expression for liver fatty acid binding protein (LFABP) by immunohistochemistry (Figure 4C). They are characterized by somatic bi-allelic inactivating mutations in the *HNF1A* (also known as TCF1) gene resulting in hepatocyte proliferation and increased lipogenesis by fatty acid synthesis and down regulation of liver-type fatty acid binding protein (LFABP), manifesting as lesional steatosis. In few cases, one mutation is somatic and the other germline. Jeannot et al. reported that heterozygous germline-inactivating mutations of the *CYP1B1* gene might increase the incidence of HCA in women with *TCF1 (HNF1A)* gene mutations [38]. Furthermore, heterozygous germline mutations in *HNF1A* are associated with an autosomal dominant condition, maturity onset diabetes of the young type 3 (MODY3) [39]. The *HNF1A* gene mutation in these patients affects only one allele and leads to a primary defect in insulin secretion by the pancreatic β cells [39,40]. Somatic inactivation of the second *HNF1A* allele confers a predisposition to develop familial liver adenomatosis in patients with MODY3 [41]. We have encountered pediatric steatotic HCAs in our practice with loss of LFABP staining but without HNF1A mutations. There is no consensus if they should be classified as h-HCA or should remain unclassifiable. Since many centers do not have advanced molecular diagnostics, these tumors may get classified as h-HCA without molecular confirmation. Loss of LFABP staining does not seem to be a reliable surrogate for HNF1A mutation. Moreover, a recent study reported 13 adults with LFABP deficient adenomas showing malignant transformation [42]. This has not been reported in children, to date.

**Beta-catenin mutated adenomas:** They are shown to be the third most common in adults and most common in the syndromic subgroup among children [26,30]. Microscopically, tumors can show focal or patchy pseudo-acinar architecture and mild cytologic atypia including small cell change (Figure 5A). Steatosis is rare and inflammation or peliosis are absent. Tumor cells are negative for SAA and CRP and show normal (retained) cytoplasmic expression of LFABP. They show nuclear and cytoplasmic reactivity for beta catenin by immunohistochemistry (Figure 5B). They also show strong and nearly diffuse staining for glutamine synthetase (Figure 5C). Glutamine synthetase is an enzyme that is involved in nitrogen metabolism and is upregulated in this subtype because of activation of GLUL, which is a b-catenin target gene that codes for glutamine synthetase. Missense mutations or deletions in *b-catenin,* seen in this group lead to activation of b-catenin [41]. The most affected exons include 3, 7, and 8 and lead to elevated nuclear and cytoplasmic levels of b-catenin [42]. While exon 7 and 8 mutations have not been associated with change in prognosis in HCA, those with exon 3 mutations have been shown to have increased risk of malignancy [43]. These tumors are the most difficult to distinguish from well-differentiated HCC and have the highest risk for malignant transformation. Patchy CD34 expression without loss of reticulin favors the diagnosis of an adenoma over a well-differentiated HCC. HCAs can arise in the setting of familial adenomatous polyposis (FAP) coli with a germline mutation the APC gene. In associating with Axin and GSK-3β, APC functions as a regulator of beta-catenin expression and localization. A bi-allelic mutation of the APC gene will disrupt this function leading to accumulation and activation of beta-catenin mimicking beta-catenin mutated adenomas. Although bi-allelic mutations of APC have been reported in HCAs arising in FAP [44], that is not necessarily true with one report of bi-allelic HNF1A mutations in a HCA arising in a patient with FAP [45,46]. Both are reports from adults and there are no data on the spectrum of somatic mutations or consensus on management strategies for FAP-related pediatric HCAs.

**Unclassified hepatic adenomas:** This category is created for tumors that lack characteristic features of any one subtype described above. Overall, they constitute 5–10% of HCAs. Some tumors that show near total necrosis, hemorrhage, or other secondary changes can be classified under this category.

**Well-differentiated hepatocellular tumors:** These are HCAs with focal atypical morphologic features such as small cell change, pseudo acinar architecture, cytologic atypia, focally thick hepatic cords and/or focal reticulin loss and may not unequivocally satisfy the diagnostic criteria for HCC. They are typically seen in adults and called “atypical hepatocellular neoplasms” [43]. It is important to note that disagreements exist concerning the distinction of adenomas from well-differentiated hepatocellular tumors. In our practice, we have seen reliance on certain histologic features more than others. National or international efforts with central review are warranted to thoroughly evaluate the histology of pediatric HCAs with respect to risk factors and outcomes.

## 6. Management, Complications and Prognosis

Management of pediatric HCAs depends on underlying medical condition, surgical candidacy, symptoms, size and number of lesions, and histopathologic subtype. Most centers begin by ruling out underlying liver disease or other medical conditions. Literature also supports management based on sex, with less aggressive approach in females than males given that males tend to have a higher risk for malignant transformation [1,7,15]. Similar aggressive approach with either embolization or resection, is employed for tumors that are larger than 5.0 cm or those with i-HCA subtype due to multiple studies demonstrating hemorrhage and spontaneous rupture in 11–29% of patients in this category [1,2,5,7,9]. n children, symptomatic tumors are usually managed more aggressively than those diagnosed incidentally [1,23]. The majority of patients with smaller HCAs (less than 5.0 cm in diameter) can be managed with discontinuation of any drug that may be driving hormone dysfunction, weight loss if the patient is overweight, and continued monitoring with MRI [1,2,5,7,9]. Given the rarity of the disease, clear guidelines do not exist and management is reliant on individual institutional experience and multidisciplinary team evaluation [43]. Multifocal tumors can be particularly challenging. Some studies have proposed resection of the larger lesions with close monitoring of the others [47]. However, there is potential for rapid growth of other tumors after resection. In adults, liver transplantation is reserved for multiple adenomas in men, or multiple large i-HCAs (any gender), or multiple beta-catenin mutated tumors (any size, any gender). Pediatric management is often based on adult guidelines and resection versus transplantation for multifocal HCAs, particularly in those without an underlying liver disease, remains an area of research for children. Multi-institutional and/or international collaborations can help create consensus guidelines for the management of these rare tumors in children.

A subset of children with ruptured HCAs that present with peritonitis or hypotension require emergent resection, embolization or other temporizing measures. While these patients can be bridged safely, the outcomes are not well reported. In case reports, complications for ruptured pediatric HCAs include bile leaks, abscess formation, and respiratory complications [48]. In contrast, non-ruptured pediatric HCAs have been reported to do well post-operatively [35,36]. Nonetheless we recognize that all liver resections are high risk procedures and thus possible complications should be thoroughly discussed with the patient and family during counseling.

While pediatric HCAs are typically solitary lesions, up to 30% of patients can present with multiple tumors [14,49]. Hepatic adenomatosis is defined as more than 10 adenomas in the liver. Liver transplantation has been proposed and attempted for hepatic adenomatosis in some centers, with positive outcomes [6,14,24]. Long-term prognosis for the majority of these post-transplant patients has not been well established but published small case reports show positive results [8,50,51]. For patients with liver adenomatosis, another consideration is resecting tumors > 5.0 cm with close monitoring of the remaining tumors [26,37]. The majority of the literature regarding prognosis belongs to adults, where HCAs have shown minimal post-operative tumor complications and no recurrence [8,23,51]. Given this lack of literature, controversy remains concerning the optimal surgical management for children with multiple tumors and optimal timing of resection. Individualized management for each patient is practiced with discussions including risk of observation, resection, and liver transplantation.

## 7. Future Perspective

The advancements in the understanding of the underlying biology of HCAs have primarily come from adult patients [9,52]. In a multi-institutional study of 533 adenomas arising in 411 patients, a new molecular subtype of HCA with activation of sonic hedgehog pathway was identified in 4% of total patients (*n* = 16, all females) with an age range of 29 to 48 years. There were no recurrent mutations in this subgroup. The subgroup was identified based on the overexpression of 6 genes (*PTGDS*, *HHIP*, *FRCLA*, *PTCH1*, *GPR97 and TNNC1*) using unsupervised hierarchical clustering. There were no pediatric patients in this subgroup and this subgroup showed significantly higher rate of symptomatic bleeding compared to others.

Advancements in subclassification have helped guide aggressive management for beta-catenin mutated adenomas and a cautious approach to manage bleeding in inflammatory and sonic hedgehog activated subtypes ^9^ Given the distinct presentation and outcomes in children, the molecular characterization of pediatric HCAs is likely unique. Likewise, the limited biological understanding of pediatric HCAs has impeded any translation to providing targeted therapies. One of the few clinical advancements in the last 20 years has been the implementation of transarterial embolization, allowing some patients to be temporized to permit surgical intervention in a less acute fashion. Nonetheless, the lack of targeted therapies has resulted in several patients requiring a partial hepatectomy, a high-risk operation. Further understanding of the tumor biology, key pathways, and overall tumor progression is warranted to create small molecule inhibitors that can help shrink or absolve these tumors.

## 8. Conclusions

In summary, HCAs are rare benign liver tumors that have been historically linked to hormonal imbalance in adults and can be seen arising in abnormal livers in children. Molecular subclassification of adult HCAs has laid the foundation for our understanding of pediatric HCAs. Despite advances in molecular subtyping, progress in treatment and risk stratification in children is limited. Additional collaborative efforts are warranted for better diagnosis, classification, and management of these rare pediatric tumors.

## Figures and Tables

**Figure 1 cancers-15-04790-f001:**
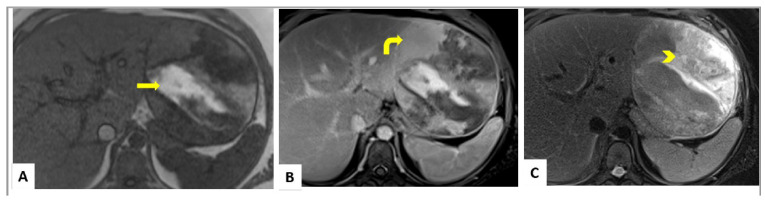
MRI images of pediatric hepatocellular adenoma. (**A**) Axial T1-weighted MRI sequence showing a discrete mass within the left hepatic lobe with regions of hyper-intense signal compatible with internal hemorrhage (yellow arrow). (**B**) Axial post-contrast T1-weighted MRI sequence showing a discrete round mass within the left hepatic lobe with enhancing solid components (curved yellow arrow) (**C**) Axial fat suppressed T2-weighted MRI sequence showing a large discrete round mass within the left hepatic lobe with heterogeneous signal intensity within (yellow arrowhead).

**Figure 2 cancers-15-04790-f002:**
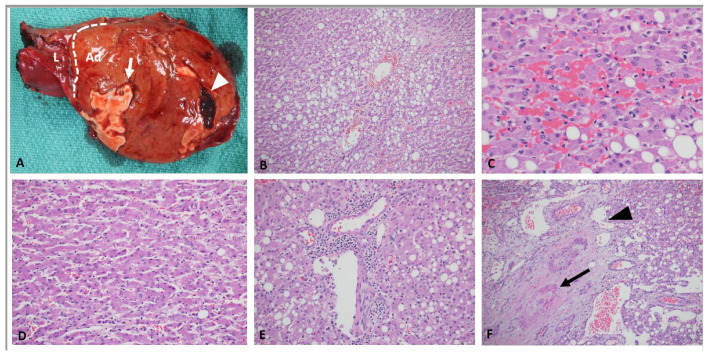
H&E of pediatric hepatocellular adenoma. (**A**) A large 17.0 cm adenoma (Ad) is seen involving the left lobe (L). Edge of the tumor is highlighted by a white dotted line. White arrow shows a geographic area of infarction with peripheral hyperemia. White triangle indicates intra-tumoral acute hemorrhage. Cut surface of the tumor is otherwise uniform, brown, and fleshy. (**B**) Adenomas, particularly those with HNF1A mutations can be often rich in fat. (**C**) Atrophic cords can be seen focally of diffusely throughout the lesion (**D**). Lesional inflammatory cells are typically seen in and around the perivascular connective tissue. (**E**) Inflammatory adenomas can show areas of sinusoidal dilatation and congestion reminiscent of peliosis. (**F**) Lesional vessels can vary from thick muscular arteries (black arrow) to dilated thin walled venous or capillary channels (black triangle). All microscopic images, H&E stained and obtained at 200× magnification.

**Figure 3 cancers-15-04790-f003:**
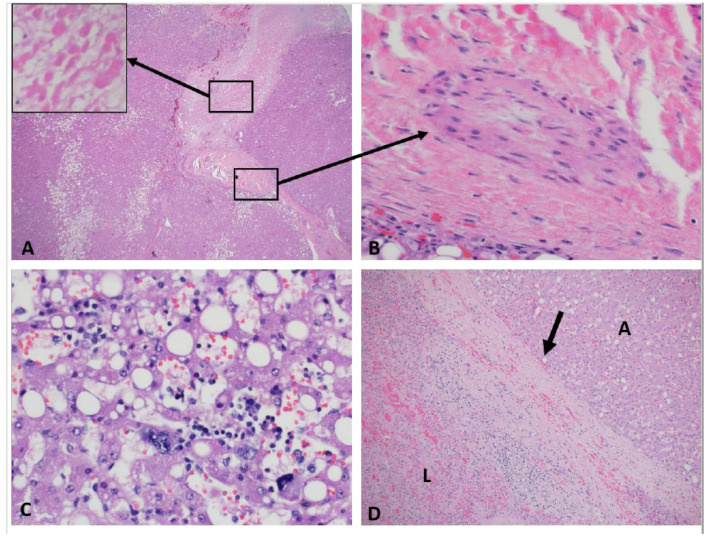
Histological characteristics seen in pediatric hepatocellular adenoma. (**A**,**B**) Some adenomas can show secondary changes such as hemorrhage and ischemia. Adenoma with focal ischemic necrosis (highlighted in the inset) is shown following a vessel showing obliterative vasculopathy. Such abnormal vessels can be seen in large, long-standing adenomas. (**C**) Nuclei can rarely show bizarre atypia but are not accompanied by mitoses and do not indicate aggressive biologic behavior. (**D**) A large, long-standing adenoma (A) can show a thickened capsule (black arrow) demarcating it from adjacent liver (L). All images H&E stained.

**Figure 4 cancers-15-04790-f004:**
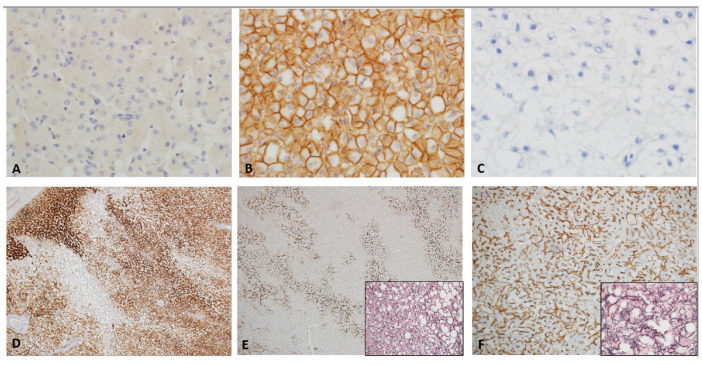
Immunostaining of pediatric hepatocellular adenoma. (**A**) Adenomas are typically diffusely negative for Glypican-3. (**B**) Beta catenin shows membranous staining in all subtypes of adenomas except in those with CTNNB1 mutations. (**C**) HNF1-alpha mutated adenomas typically show loss of liver fatty acid binding protein (LFABP) expression. (**D**) Hepar generally shows strong and diffuse cytoplasmic staining with some tumors showing focally negative areas. (**D**) CD34 highlights blood vessels traversing the tumor and sinusoids adjacent to the vascular connective tissue. Inset shows preserved reticulin within the tumor with widened cords in steatotic areas. (**F**) Image in panel F is the same tumor as panel (**E**). This adenoma showed a focal area of diffuse CD34 expression; however, reticulin was preserved in this area helping to differentiate from a well–differentiated HCC. Such areas can pose diagnostic challenge on small biopsies.

**Figure 5 cancers-15-04790-f005:**
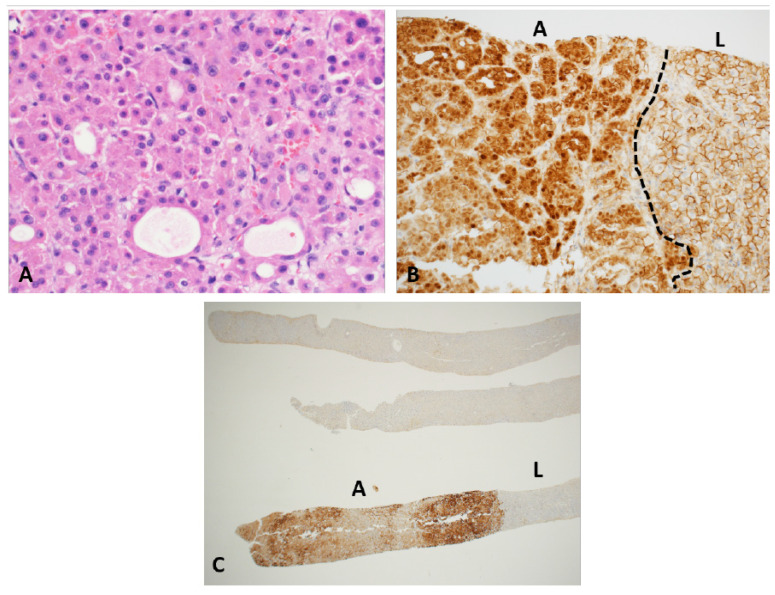
Beta catenin and glutamine synthetase of pediatric hepatocellular adenoma. (**A**) Beta catenin mutated adenoma showing focal pseudo-acinar architecture. (**B**) Immunostaining for beta catenin shows nuclear and cytoplasmic staining in the lesional cells (A) with membranous staining in the adjacent non-tumor liver (L), separated by black dotted line. (**C**) Glutamine synthetase shows diffuse and strong cytoplasmic staining in the adenoma and negative staining in the adjacent non-tumor liver.

**Table 1 cancers-15-04790-t001:** Differences between adult and pediatric HCA.

Characteristics	Pediatric HCA	Adult HCA
Background Liver	Often abnormal	Usually normal
Incidence	1:1,000,000	4:100,000
Risk factors	Genetic disorders, underlying hepatic parenchymal diseases, obesity	Obesity, alcoholism
Sex predisposition	Equal incidence in pre-pubertal girls and boys but higher in post-pubertal girls.	Females > Males
Hormonal abnormality	Elevated estrogen, exposure to anabolic steroids.	Elevated estrogen, exposure to oral contraceptive pills and anabolic steroids

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
