# Peer review of "Pediatric Hepatocellular Adenomas: What Is Known and What Is New?"

_cancers, 2023, doi:10.3390/cancers15194790_

Round 1

Reviewer 1 Report

PediatricHCA is, as written by Dr Espinoza and colleagues, a rare liver tumor. Therefore, this concise review is a valuable kind of "mini state of the art". It is well written, clear and well documented. 

I only have few minor remarks , specially for HCAs related to glycogen storage disease.

In paragraph 2, lines 50 and 51,glycogen storage disease (GSD) should read glycogen storage diseases. There are 15 sub-types  of GSD , and only 2 of them, i.e type I and III (type I being involved for the majority of cases) are likely to be associated with the development of HCA. Maybe, the subtypes could be added. The same remark concerns line 149 in paragraph 5

It is very true that, clear guidelines do not exist  (paragraph 6, lines 269 to 271), owing to the rarity of the disease. If most teams consider 5 cm as the threshold for chosing surgical resection, this is not always the case in patients with GSDs who often have multiple adenomas . It should be mentioned that liver resection is likely to accelerate the development of remaining tumors on the one hand, and to complicate liver transplantation, should it be later .indicated 

Author Response

In paragraph 2, lines 50 and 51,glycogen storage disease (GSD) should read glycogen storage diseases. There are 15 sub-types  of GSD , and only 2 of them, i.e type I and III (type I being involved for the majority of cases) are likely to be associated with the development of HCA. Maybe, the subtypes could be added. The same remark concerns line 149 in paragraph 5

Thank you for your comment. We have edited this portion to state “glycogen storage diseases” and have included that types I and III are involved in most of the cases that are associated with the development of HCA in both paragraph 2 and paragraph 5.

It is very true that, clear guidelines do not exist (paragraph 6, lines 269 to 271), owing to the rarity of the disease. If most teams consider 5 cm as the threshold for chosing surgical resection, this is not always the case in patients with GSDs who often have multiple adenomas . It should be mentioned that liver resection is likely to accelerate the development of remaining tumors on the one hand, and to complicate liver transplantation, should it be later .indicated .

Thank you for your comment. We have addressed this by stating “Multifocal tumors can be particularly challenging. Some studies have proposed resection of the larger lesions with close monitoring of the others. However, there is a potential for rapid growth of other tumors after resection. In adults, liver transplantation is reserved for multiple adenomas in men, or multiple large i-HCAs (any gender), or multiple beta-catenin mutated tumors (any size, any gender). Pediatric management is often based on these adult guidelines and resection versus transplantation for multifocal HCAs, particularly in those without an underlying liver disease, remains an area of research for children.”

Reviewer 2 Report

This review article provides an overview of the epidemiology, risk factors, clinical features, imaging characteristics, pathology, subclassification, and management of hepatocellular adenomas (HCAs) in children. HCAs are rare benign liver tumors in children, often associated with genetic disorders or abnormal background livers unlike in adults where they arise in normal livers. Attempts to apply the adult HCA classification system of inflammatory, HNF-1α mutated, β-catenin mutated, and unclassified subtypes to pediatric HCAs have limitations, with many pediatric cases being unclassified. Small asymptomatic HCAs can be monitored while resection is recommended for larger lesions >5cm to prevent complications like hemorrhage. Molecular characterization of pediatric HCAs is needed to better understand biology and guide management.

=This review articale has several strengths:

-Comprehensive overview of the epidemiology, risk factors, clinical presentation, imaging, pathology, subclassification, and management of pediatric HCAs.

-Discusses the differences between HCAs in children compared to adults, such as the association with genetic disorders and abnormal background livers in children.

-Reviews the molecular subclassification system for HCAs based on adult literature and notes limitations in applying this to pediatric HCAs.

-Provides recommendations for diagnosis, surveillance, and treatment based on tumor size and other factors.

-Includes figures demonstrating histopathology and radiologic images to aid understanding. 

=However, serval weaknesses need to be concisely addressed:

        Does not critically analyze the available literature on pediatric HCAs, which is limited given the rarity of these tumors. Relies heavily on extrapolating from adult data.

        Does not mention controversies or disagreements among experts regarding diagnosis, classification, or management of pediatric HCAs.

        Provides recommendations but does not grade them based on level of evidence.

        Does not highlight significant knowledge gaps or unanswered questions in the pediatric HCA literature to inform future research.

        Does not discuss in detail the rationale behind size cut-offs for treatment versus observation.

        Overall this is a well-written focused review on pediatric HCAs despite limited data. A more critical analysis of the literature, controversies, and evidence gaps would further strengthen the review.

none.

Author Response

Does not critically analyze the available literature on pediatric HCAs, which is limited given the rarity of these tumors. Relies heavily on extrapolating from adult data.

Thank you for your comment. We have re-reviewed the pediatric literature and added specific information in several sections as follows:

  1. Risk Factors and Epidemiology: added several genetic and acquired conditions in which HCAs have been described in children.
  2. Presentation and Clinical Features: In contrast to adults, a significant subset of pediatric HCAs occur in the context of genetic disorders or other acquired conditions. Hence, they are often diagnosed incidentally during surveillance.
  3. Imaging: We added a paragraph on the use of CT in the diagnosis of pediatric HCAs.
  4. Histopathology/Genetics: We have encountered pediatric steatotic HCAs in our practice with loss of LFABP staining but without HNF1A mutations. There is no consensus if they should be classified as h-HCA or should remain unclassifiable. Since many centers do not have advanced molecular diagnostics, these tumors may get classified as h-HCA without molecular confirmation. Loss of LFABP does not seem to be a reliable surrogate for HNF1A mutation. A recent study reported 13 adults with LFABP deficient adenomas showing malignant transformation. This has not been reported in children, to date.

Does not mention controversies or disagreements among experts regarding diagnosis, classification, or management of pediatric HCAs.

Added specific sentences such as:

  1. There is no consensus on the requirement of SAA or CRP positivity or a specific mutation to diagnose an inflammatory HCA. Most cases are diagnosed on morphology alone. Cases without an inflammatory component but with positive staining for SAA or CRP are particularly challenging and remain unexplained.
  1. HCAs can arise in the setting of familial adenomatous polyposis (FAP) coli with a germline mutation the APC gene. In associating with Axin and GSK-3β, APC functions as a regulator of beta-catenin expression and localization. A bi-allelic mutation of the APC gene will disrupt this function leading to accumulation and activation of beta-catenin mimicking beta-catenin mutated adenomas. Although bi-allelic mutations of APC have been reported in HCAs arising in FAP, that is not necessarily true with one report of bi-allelic HNF1A mutations in a HCA arising in a patient with FAP. Both are reports from adults and there is no data on the spectrum of somatic mutations or consensus on management strategies for FAP related pediatric HCAs. 

Provides recommendations but does not grade them based on level of evidence.

We agree that we have not explicitly graded the recommendations based on the level of evidence but when relevant we have elaborated on the quality of evidence stating the number of patients involved, their age (adult vs pediatric) etc. This is a limitation of our review given the lack of clear guidelines for pediatric HCAs

Does not highlight significant knowledge gaps or unanswered questions in the pediatric HCA literature to inform future research.

Added sentences such as:

Multifocal tumors can be particularly challenging. Some studies have proposed resection of the larger lesions with close monitoring of the others. However, there is a potential for rapid growth of other tumors after resection. In adults, liver transplantation is reserved for multiple adenomas in men, or multiple large i-HCAs (any gender), or multiple beta-catenin mutated tumors (any size, any gender). Pediatric management is often based on adult guidelines and resection versus transplantation for multifocal HCAs, particularly in those without an underlying liver disease, remains an area of research for children.  

Does not discuss in detail the rationale behind size cut-offs for treatment versus observation.

We have added a sentence to elaborate on this:

Aggressive approach with either embolization or resection, is employed for tumors that are larger than 5.0 cm or those with i-HCA subtype due to multiple studies demonstrating hemorrhage and spontaneous rupture in 11-29% of patients in this category [1,2,5,7,9].In children, symptomatic tumors are usually managed more aggressively than those diagnosed incidentally [1,15]. The majority of patients with smaller HCAs (less than 5.0 cm in diameter) can be managed with discontinuation of any drug that may be driving hormone dysfunction, weight loss if the patient is overweight, and continued monitoring with MRI [1,2,5,7,9]. Given the rarity of the disease, clear guidelines do not exist and management is reliant on individual institutional experience and multidisciplinary team evaluation.

Overall this is a well-written focused review on pediatric HCAs despite limited data. A more critical analysis of the literature, controversies, and evidence gaps would further strengthen the review.

Thank you for your thorough review and analysis. 

Reviewer 3 Report

I have carefully reviewed the article you sent me. Here are my suggestions:

In my opinion, there is no need for the phrase "A concise review of literature" in the article title.

Instead of the expression "greater than 10 adenomas in the liver" in the article text, the expression "more than 10 adenomas in the liver" may be more appropriate.

There is no need to use the phrase "gastroenterologists" in the summary section. Because the term "hepatologist" is already used in the text of the article. Hepatology is a specialized sub-branch of gastroenterology.

Under the Imaging subheading, sufficient information is given about ultrasonography AND MRI, but the subject of computed tomography is not mentioned at all. The place of computerized tomography (especially dynamic liver tomography), which is the most commonly used in the world, should be included in the diagnosis of hepatic adonoma. Additionally, a few sentences should be written about contrast ultrasonography.

Author Response

In my opinion, there is no need for the phrase "A concise review of literature" in the article title.

We have eliminated this phrase from the article title.

Instead of the expression "greater than 10 adenomas in the liver" in the article text, the expression "more than 10 adenomas in the liver" may be more appropriate.

We have replaced the phrase with “more than 10 adenomas in the liver”

There is no need to use the phrase "gastroenterologists" in the summary section. Because the term "hepatologist" is already used in the text of the article. Hepatology is a specialized sub-branch of gastroenterology.

Thank you for this comment, we have eliminated “gastroenterologists.”

Under the Imaging subheading, sufficient information is given about ultrasonography AND MRI, but the subject of computed tomography is not mentioned at all. The place of computerized tomography (especially dynamic liver tomography), which is the most commonly used in the world, should be included in the diagnosis of hepatic adonoma. Additionally, a few sentences should be written about contrast ultrasonography.

To address the comment concerning lack of description of computed tomography we have written: While US and MRI are the currently recommended imaging modalities for detection and monitoring of pediatric HCA, we recognize that several institutions worldwide may still rely on computed tomography (CT). The utility of CT is valid in situations where there is a lack of other imaging tools. Regardless, MRI and US should be utilized when available. The downside with CT is the high radiation dose imparted, especially when multiphase CT imaging is performed for the evaluation of suspected hepatic adenoma. On CT, HCA appear without central scar and can show a heterogenous appearance given hemorrhage in the tumor [13] Also, multi focal tumor burden may be hard to assess given the poor soft tissue contrast resolution, especially when a single post contrast phase is obtained. 

We have addressed your comment by including the following about contrast enhanced ultrasonography: Recent studies have found that contrast enhanced US can help differentiating an HCA from focal nodular hyperplasia (FNH), as HCAs show centripetal arterial flow While some evidence exists that contrast enhance US has similar specificity as contrast enhanced MRI, other have demonstrated that the use of contrast enhanced US can be used as an adjunct improve the sensitivity of differentiating HCA from other liver masses.”

Reviewer 4 Report

I have read with interest this manuscript, which appears to be overall well-written and easy to be read. It concerns a narrative review on a rare topic, which anyway it is interesting. I would suggest to add a table to make a more easily and fast lecture of those main features that appears to be distinctive of pediatric HAs with the intent ot underline similitudines and differences with adult HAs.

Author Response

I have read with interest this manuscript, which appears to be overall well-written and easy to be read. It concerns a narrative review on a rare topic, which anyway it is interesting. I would suggest to add a table to make a more easily and fast lecture of those main features that appears to be distinctive of pediatric HAs with the intent to underline similarities and differences with adult HAs.

Thank you for your comments. We have included the following summary table into the paper to illustrate this important point you mention.

Characteristics

Pediatric HCA

Adult HCA

Background Liver

Often abnormal

Usually normal

Incidence

1:1,000,000

4:100,000

Risk Factors

Genetic disorders, underlying parenchymal diseases, obesity

Obesity, alcoholism

Sex

Equal incidence in prepubertal girls and boys but higher in post pubertal girls.

Females > Males

Hormonal Abnormality

Elevated Estrogen, exposure to anabolic steroids.

Elevated Estrogen, exposure to oral contraceptive pills and anabolic steroids.

This material is original research, has not been previously published, and is not currently being considered for publication elsewhere, and I am attesting to these facts on behalf of all authors of this manuscript. Again, we would like to thank all of the reviewers as well as all editors for the opportunity to publish our work in the Cancers.